# Development of Biotechnological Photosensitizers for Photodynamic Therapy: Cancer Research and Treatment—From Benchtop to Clinical Practice

**DOI:** 10.3390/molecules27206848

**Published:** 2022-10-13

**Authors:** Mariza Aires-Fernandes, Ramon Botelho Costa, Stéphanie Rochetti do Amaral, Cassamo Ussemane Mussagy, Valéria C. Santos-Ebinuma, Fernando Lucas Primo

**Affiliations:** 1Department of Bioprocess and Biotechnology Engineering, School of Pharmaceutical Sciences, São Paulo State University—UNESP, Araraquara 14800-903, São Paulo, Brazil; 2Escuela de Agronomía, Facultad de Ciencias Agronómicas y de los Alimentos, Pontificia Universidad Católica de Valparaíso, Quillota 2260000, Chile

**Keywords:** cancer therapy, photodynamic therapy, biotechnological photosensitizers, nanobiotechnology, malignant tumors

## Abstract

Photodynamic therapy (PDT) is a noninvasive therapeutic approach that has been applied in studies for the treatment of various diseases. In this context, PDT has been suggested as a new therapy or adjuvant therapy to traditional cancer therapy. The mode of action of PDT consists of the generation of singlet oxygen (¹O_2_) and reactive oxygen species (ROS) through the administration of a compound called photosensitizer (PS), a light source, and molecular oxygen (^3^O_2_). This combination generates controlled photochemical reactions (photodynamic mechanisms) that produce ROS, such as singlet oxygen (¹O_2_), which can induce apoptosis and/or cell death induced by necrosis, degeneration of the tumor vasculature, stimulation of the antitumor immune response, and induction of inflammatory reactions in the illuminated region. However, the traditional compounds used in PDT limit its application. In this context, compounds of biotechnological origin with photosensitizing activity in association with nanotechnology are being used in PDT, aiming at its application in several types of cancer but with less toxicity toward neighboring tissues and better absorption of light for more aggressive types of cancer. In this review, we present studies involving innovatively developed PS that aimed to improve the efficiency of PDT in cancer treatment. Specifically, we focused on the clinical translation and application of PS of natural origin on cancer.

## 1. Introduction

In 1900, Professor Herman Von Tappeiner and Oscar Raab conducted studies that contributed to the subsequent application of many photodirected approaches, since they evidenced the influence of light in one of their experiments in which they noticed the occurrence of photoactivation of the acridine dye during parametric treatment, culminating in cell death [1,2]. This achievement brought to medicine a new perspective regarding photosensitizers, since the optical properties of photoactivation of some compounds, in order to generate a therapeutic or diagnostic response through the emission of light, were not known. Thus, based on these new principles, many concepts and techniques that use photosensitizers have evolved, enabling the formation of theranostic agents (agents that have the ability to generate an image and therapeutic effect when excited with light), as well as the use of practices such as photodynamic therapy (PDT) [2,3].

According to Sandland et al., the study of photosensitive theranostic agents has provided interesting resources for the concept of personalized medicine, which is based on the precepts that patients have different needs, complications, and responses, which consequently end up demanding personalized solutions for the effective combat of their respective diseases. Thus, theranostic agents have been studied as noninvasive compounds for the personalized diagnosis of diseases such as cancer, since they are able to quantify them through the combination of this agent with positron emission tomography (PET) equipment and combat them through the therapeutic effects provided by photoactivation [3].

The applicability of PDT and the modulation of its therapeutic effects are closely related to the presence of molecular oxygen, predilection for the best photosensitizer (PS), and standardization of the specific wavelength of light, since the technique is mainly based on the generation of reactive oxygen species (ROS) through photochemical reactions induced by the PS that absorbed the light at a certain wavelength, thus modulating a cytotoxic effect from which there is the accumulation/delivery of this photosensitive component [4].

In this sense, the selection of PS is extremely important, since it is desired to maximize the therapeutic effect without tissue damage. For this, an important characteristic of PS is highlighted, being the characteristic wavelength of which its respective photoactivation occurs, and this length must be able to penetrate the target tissue, maintaining its intensity to excite the PS, interacting minimally with other cells and tissues. Molecules such as myoglobin and hemoglobin are able to absorb part of the light, depending on the wavelength used, as well as melanin, which ends up absorbed in the spectrum between 500 and 600 nm. Thus, the best PS are those that have an absorption spectrum above 600 nm in a region between 600 and 800 nm, which results in a range beyond which the emitted light will cause minimal damage to healthy tissue by minimal interactions, ensuring that the therapy remains minimally invasive and provides destruction of the target from which PS has accumulated. Other equally important points during the screening of a PS include analyzing the parameters related to the pharmacokinetics and pharmacodynamics of the PS in question, as these characteristics culminate in factors correlated with the bioavailability of PS in the body, which is a crucial point for understanding issues such as toxicity, therapeutic dose, and agent solubility [3,4]. 

However, among the different photochemical reactions, those dependent on oxygen can be classified as type I and II reactions, both of which promote the exit of PS from its fundamental state to the excited triplet state when it is photoactivated but differ in the specimens of radicals generated, as well as in the set of interactions that lead to their formation in each type [5].

Several effects can be modulated through the reactive species produced for an effective fight against various types of cancers, such as the cytotoxicity (the accumulation of ROS in target cells are able to lead to cell death by apoptosis and necrosis) and the recruitment of the immune system through the regulation of the biosynthesis of compounds such as cytokine and hypoxia through the destruction of blood vessels (Figure 1). However, it is worth noting that this sophisticated set of effects promoted by the technique is not limited only to oncological applications, existing studies indicate the possibility of the application of PDT for the treatment of diseases such as atherosclerosis, Alzheimer’s, gastrointestinal, dermatological, and ophthalmological. The range of possibilities is so wide that even applications of the therapy can be adapted so that there is a treatment through the inactivation of different microorganisms, leading the technique to receive another name, the “photodynamic inactivation of microorganisms”, a technique that has shown possibilities of application to COVID-19 [5,6,7].

The existence of strategies to maximize the therapeutic effect are well-known, and among them, the adoption of drug delivery systems stands out, since they can be composed of different arrangements and various components in order to generate an efficient and targeted delivery system that increases the bioavailability profiles of PS, as this concept is a widely used approach in many branches of science [8,9]. An interesting implementation of this approach is the process of coating nanoparticulate systems with components originating from cell membranes in an attempt to use biomimicry on their biological designs. The advantage of this procedure would be to take advantage of the different biological interfaces existing in the in vivo model, allowing more dynamics in the therapeutic approach, such as even more effective vectorization and immune escape. In this way, phototherapy can use these coated nanoparticles for better targeting to tumor cells, of which studies have pointed out some coating possibilities for targeted delivery [10].

However, much of the evolution of PDT is due to the constant search for better photosensitizers, which, according to their discovery and the screening process, led to the categorization of PS into three different categories or generations. These take into account phototoxic potential factors, since, with higher quantum yields of singlet oxygen, greater cytotoxicity will be caused, minimal activity without photoactivation, and light absorption peaks suitable for minimally invasive therapy, amongst others. Thus, the second generation surpasses the first by having photosensitizers with these more refined biochemical aspects, and the third generation surpasses the second due to the better targeting of these molecules, as well as their greater solubility due to their association with carrier systems [6,11].

Currently, there are already studies aimed at obtaining photosensitizers by mutagenic processes, such as the studies promoted by Gorgachev et al., which were conducted with the intention of generating a monomeric photosensitizer through mutagenesis of the red fluorescent protein, KillerRed, thus obtaining the photosensitizer called SuperNova. However, they evaluated that the phototoxic potential of the PS obtained was far below the ideal, as well as discovered the extreme difficulty associated with presenting a better performance for PS in terms of phototoxic potential, either by mutagenesis or rational design [12]. 

For the production of genetically encoded photosensitizers, it is extremely important to have a broad knowledge about the structural arrangement and its respective functional determinants, so that one can build a design that provides the ideal conformation for a given purpose, i.e., aspects such as the optimized position of a chromophore (aromatic portion responsible for the main absorption of photons during photoactivation), so it can contribute to a higher production of ROS when these are positioned exogenously and so that some may or may not require cofactors for a better stimulation. Therefore, studies aimed at forming more and better selection methods, as well as a better understanding about the structural aspects and their effects during a photoactivation process, are important to advance the formation of better photosensitizers [11,12].

## 2. Photosensitizers Compounds

Photosensitizers are compounds that induce photochemical or photophysical processes when exposed to light that is in their spectral range of absorption, this being a key component of PDT [13,14]. Generally, light with a long wavelength is chosen, because it will have enough energy to start the formation of singlet oxygen [13]. The first PDT sensitizer to receive FDA (Federal Drug Administration) approval, Photofrin™, was introduced to the market in 1995 and demonstrated encouraging results in patients with high-grade dysplasia associated with Barrett’s esophagus, esophageal, and endobronchial cancer [15]. Although new at the time in this therapeutic field, this hematoporphyrin medication had several shortcomings, including chemical heterogeneity and limited penetration [15]. 

As photosensitizers have progressed from the first generation to modern ones used for PDT of various diseases, certain features of a photosensitizer suitable for PDT have been specified [16]. In this respect, the complex natural combination of oligomeric hematoporphyrin derivatives, known as HpD, is regarded as a member of the first-generation PS [17]. In order to treat bladder cancer, PDT based on a HpD was employed in the 1970s [18]. The first-generation PSs were successfully used to treat a variety of malignant tumors, according to Dougherty et al. Despite the first-generation PSs’ success, they have a number of limitations that prevent them from being used in clinical settings [18]. These limitations include the difficulty of synthesizing and purifying HpD, the excitation wavelength’s shortness, the light’s low absorption rate, and HpD’s long half-life, which makes it easy to cause unwanted phototoxicity [18]. 

The second-generation PS are synthetic substances made from or derived from porphyrins, bacteriochlorins, phthalocyanines, chlorins, benzoporphyrins, curcumin, and derivatives of methylene blue, among other substances [16]. Second-generation PS have a number of advantages over first-generation, including longer absorption wavelengths, the ability to be activated by near-infrared (NIR) light, greater depth of effect, high singlet oxygen quantum yield, an better tissue selectivity, as well as the ability to be quickly metabolized, which reduces the side effects [18]. Third-generation PS is distinguished by the combination of second-generation PS with specific components such as antibodies, carbohydrates, amino acids, or peptides or by encapsulating into carriers such as liposomes, micelles, and nanoparticles to increase PS accumulation at the specific targeted sites [15]. Multiple nanomaterials that can function as nano-PSs or PS carriers have emerged as a result of the rapid ongoing development of nanotechnology [18]. Currently, there are frequently used clinically approved PSs, which are (1) HpD, (2) hematoporphyrin derivative—porfimer sodium, (3) 5-aminolevulinic acid (5-ALA), and (4) m-tetrahydroxyphenylchlorine (mTHPC) [19]. Some of the drawbacks of using these PSs in PDT is their high degree of phototoxicity, which is brought on by their slow systemic clearance periods and high levels of PS retention [19].

## 3. Compounds from Microbial Origin—Biotechnological Photosensitizers

Natural-based biomolecules have already been used since ancient times to treat and prevent several diseases [20,21]. However, the proliferation of eco-friendly procedures and the rapid development of biotechnology have shown that it is possible to obtain outstanding photoactivatable secondary metabolites without the need of a massive plant collection [22,23,24]. In nature, microorganisms such as yeast, bacteria, and microalgae are the most common potential sources to produce certain types of compounds, including natural colorants [23]. These compounds, which are a group of chemically heterogeneous and biosynthetically unrelated molecules, present in their electronic structure a chromophore responsible for their color characteristic and due to their chemical structure can be used as an alternative to photosensitizers (PSs) from synthetic origins. Moreover, generally natural colorants are less toxic and more biocompatible than the synthetic counterparts [23,25]. The advantages of microorganisms over plant cells are their metabolic versatility: no seasonal limitation for PS production, easy cultivation, the use of carbon and nitrogen sources, and low nutritional needs for the production of bioactive molecules [20,21,26,27].

However, there is evidence that, in the last few years, little attention has been paid to the process (upstream and downstream) to obtain microbial-based PSs for application in PDT. In fact, the lack of studies makes the microbial-based PS implementation on the market more challenging. For the application of new microbial-based PSs in PDT, certain particular conditions are required, viz., (i) low dark toxicity, (ii) high-purity, (iii) high-coefficient of absorption (600–800 nm), (iv) high activation capacity and efficient reaction with light to generate singlet oxygen, (v) avoidance of overlapping of PS absorption bands with some endogenous pigments such as melanin or hemoglobin, and (vi) easy excretion capacity in order to avoid phototoxicity, among others [28,29]. Furthermore, the affinity/solubility of PSs to aqueous solutions should be carefully evaluated, as some hydrophobic PSs have revealed a high aggregation in water, hindering the in vivo utilization [30,31]. To overcome these issues, some authors have previously incorporated hydrophobic PSs in liposomes, emulsions, or nanoparticles to improve the solubility in aqueous solutions [32,33].

To our knowledge, there are several ongoing studies from different research groups examining the efficacy of microbial-based PSs for their application in PDT; however, to date, there have been few reported studies. Barnhart-Dailey et al. [34] reported the use of tolyporphin with a strong phototoxicity in PDT obtained from cyanobacteria. Following the same line, Gomaa et al. [35] revealed the PS potential of chlorophyllin (chlorophyll derivative) produced by cyanobacteria. In this particular case, the authors also revealed the positive optical properties (600–670 nm) and the great affinity of chlorophyllin in aqueous solutions. We assume that the main problem of PS production from microbial sources is related to the process cost. The process is composed of two main steps: (i) upstream—molecule production by the microorganism in orbital shaker or bioreactor (more interesting from an industrial point of view)–and (ii) downstream—molecule recovery from the biomass (intracellular compounds) or extracellular media. In the upstream step, high amounts of molecules and an optimization of the processing parameters such as aeration, time, temperature, and culture medium are required for the improvement of microbial-based biomolecule production [21,36]. On the other hand, in the downstream step, several unit operations to achieve the biomolecule purity are necessary for its application as PS [37,38]. In this sense, the final cost of microbial PSs is higher than their vegetable counterparts. It is important to emphasize that, at this stage, the adequate selection of more benign, green, and safe extractant and polishing agents is of paramount importance considering the final application of the biomolecule (cf., PSs) [39,40].

Current trends in the use of natural products are promoting the search for new microbial-based compounds, including PSs (Figure 2). For instance, the natural PSs such as: curcumins [41,42], tolyporphin A [43], chlorophyllin [44], riboflavin [45], and anthraquinones [30], among others produced by plant sources as PSs, can also be obtained by microorganisms. For example, Singh et al. [44] evaluated the production of chlorophyllin from the green microalga *Chlorella minutissima*, resulting in a chlorophyllin yield of up to 65.3 mg.g^−1^. You et al. [45] using a sequential optimization strategy to enhance the production of riboflavin by recombinant *Bacillus subtilis* RH44, reached a final riboflavin concentration of 16.36 g/L obtained after 48 h of fermentation in a 5-L bioreactor. The microbial production of curcumin by recombinant *Escherichia coli* cells was revealed by Kim et al. [46], who achieved yields of curcuminoids up to ~100 mg.L^−1^. The production of red pigments (anthraquinones) from biotechnological origin was obtained using a mixture of filamentous fungi (*Talaromyces minioluteus* and *Penicillium minioluteum*) [47]. Galanie and Smolke [48] evaluated protoberberine alkaloid production using recombinant yeast (*Saccharomyces cerevisiae*). In this particular work, the authors evaluated several strategies, viz., enzyme variant screening, culture optimization, and genetic copy number variation, which led to the improvement of canadine (1.8 mg.L^−1^), facilitating the extension of the yeast pathway to biosynthesize berberine. Cercosporin is another potential PS that can be obtained using microbial-based sources (*Bacillus velezensis* and *Lysinibacillus* sp.) [49] to achieve 984.4 and 626.3 mg.L^−1^, respectively, after 4 days of coculture.

As clearly observed, most microbial-based PSs are not yet used for PDT applications, but similar to other PSs (synthetic), their use has been pointed out as the most promising. However, for the microbial production of PSs to become a reality, advances and understanding of the biosynthesis pathways for improved production and the advancement of integrated extraction/purification and polishing units are still required. The increase in the global market for natural compounds (cf., PSs) has led to the development of sustainable production of PSs for application in PDT, first to achieve higher production titers of biomolecules and second to reduce the overall processing cost.

## 4. Application of PDT Photosensitizers in Cancer Treatment

In the last decade, several in vitro studies involving the application of PDT in the treatment of cancer were developed, with the main focus on better understanding the efficacy and selectivity of photosensitizers (Table 1). The summarized research works aimed to demonstrate the results of the use of photosensitizing agents belonging to the second and third generations of PSs. This is because these groups have advantages that include greater absorption wavelengths, which consequently allow PS to reach deeper tissues, allowing for the targeting of cancer cells [50].

In this context, recently, nanoconstructs consisting of upconverting nanoparticles (UCNPs) have been frequently used to solve limitations of PSs such as low tissue penetration and low tumor selectivity. UCNP nanoconstructs are capable of transforming near-infrared (NIR) light into visible or UV light using NIR irradiation from 700 to 1000 nm and can be associated with PSs suitable for treating tumors in deeper tissues. Other advantages associated with UCNP NIR to visible-based PDT include reduced autofluorescence and high hydrophilicity and selectivity. In addition to all the advantages of the UCNPs, there is the possibility of their carrying out maintenance in the PS structure [51,69].

In the study conducted by Cui et al., the development of multifunctional UCNPs coated with folate-conjugated chitosan (FASOC) was reported. Zinc II phthalocyanine (ZnPc) (FASOC-UCNP-ZnPc) was used for PDT targeted for the in vivo treatment of breast cancer and sarcoma (Figure 3A). For this, the authors induced the tumors in female mice and evaluated the toxicity of FASOC-UCNP-ZnPc (0, 36, 51.45, 73.5, 105, 150, and 220 mg.kg^−1^) through the injection of 300 μL intravenously. For PDT treatment, the animals were divided into three groups. Each group was treated with FASOC-UCNP-ZnPc (50 mg.kg^−1^, containing 2.88 mg.kg^−1^ ZnPc) with irradiation at 980 and 660 nm (0.2 W.cm^−2^) for 30 min. They evaluated the effectiveness of the treatment through the tumor volume, body weight, and survival rate of the animals. They observed that there was no difference between the treatment of the subcutaneous tumor with light at 980 and 660 nm, as the tumor volume was equal. However, for deep tumors, the 980 nm PDT was more efficient. FASOC-UCNPs have demonstrated a safe in vivo dose of <150 mg.kg^−1^. They concluded that the nanoconstructs were effective in treating deep tumors with higher ROS generation compared to ZnPc under 660 nm light [51].

Another study involving nanoparticles targeting tumor cells was conducted by Wang et al. The nanoplatform was based on HOS (MH) cell membrane-coated PLGA/PVA (Figure 3B). A heptamethine cyanine molecule (IR780) was used as PS. IR780 can be activated under near-infrared irradiation (NIR) at 808 nm. The authors verified the constructed homologous targeting-based nanoplatform (MH-PLGA-IR780 NPs) in osteosarcoma PDT. In vitro biological assays involving osteosarcoma cell lineage and the measurement of intracellular ROS of MH-PLGA-IR780 NPs (IR780:5 μg.mL^−1^, 1.5 W.cm^−2^ for 2 min, 808 nm) were evaluated. Mice bearing osteosarcoma were used for treatment with PDT. The protocol used involved the application of 5 mg.mL^−1^ of MH-PLGA-IR780 NPs, with a laser power of 2 W.cm^−2^ during 5 min of irradiation. Subsequently, the size of the tumors and weight of the animals were evaluated every 4 days for 16 days. It was observed that MH-PLGA-IR780 NPS-PDT induced apoptosis and ferroptosis. A similar effect was evaluated in the in vivo model, as MH-PLGA-IR780 was able to almost completely inhibit tumor growth. Therefore, the obtained targeted NPs irradiated with NIR were efficient in vitro and in vivo PDT and could be used in other types of tumors [58].

Another example of nanoparticles developed with specific targeting to tumor cells was described by Tian et al. [57]. In this study, nanoparticles loaded with folate-conjugated selenium-rubirin (FA-NMe_2_Se_4_N_2_NPs) were developed for the application of PDT in the treatment of cervical carcinoma in vitro and in vivo (Figure 3C). For this, the synthesis of NP was conducted in such a way that the targeted PS was activated only at acidic pH [57]. This was due to the fact that pH-activated PS produce more ROS under irradiation. The PDT protocol used near-infrared irradiation (NIR). For the in vitro assay, cervical carcinoma cells were incubated with 50 μg.mL^−1^ of FA-NMe2Se4N2 NPs and irradiated with a 635-nm laser at a power of 100 mW.cm^−2^ for 300 s. For the in vivo assay, mice bearing the subcutaneous tumor were treated with 100 μL to 0.5 mg.mL^−1^ of FA-NMe_2_Se_4_N_2_ NPs; after eight hours, the tumor region was irradiated with an 808-nm laser at 100 mW.cm^−2^ for 30 min. The tumor size was assessed for 15 days after PDT treatment. It was observed that the in vitro phototoxicity was dose-dependent. For the in vivo experiment, they noted that irradiation at 808 nm inhibited tumor growth almost completely. Through the results obtained, they concluded that the NPs produced were selective and presented a high efficiency for the treatment of cancer [57].

Several works have used porphyrin derivatives for application in PDT in the treatment of cancer [55,56]. Lu et al. tested the nanoscale metal–organic structure combination of Hf-porphyrin DBP-UiO (DBP-UiO NMOF) as a strategy for application in PDT in the treatment of head and neck cancer. They observed that the tumor was eradicated in 50% of mice treated with a single dose of DBP-UiO and a single exposure to light [55]. Another porphyrin conjugate was described in the work by Song et al. for colon cancer PDT. The conjugate composed by the association of porphyrin with cisplatin incorporated in polymeric nanocarriers (NP@Pt-1) was tested in vitro and in vivo. They observed cell death and ROS generation capable of inducing immunogenic cell death, a promising effect for application in other types of cancer and with other chemotherapeutic agents [56].

Photofrin isolates were described in preclinical studies developed by Shi et al. and Wu et al. [63,64]. Sodium sinoporphyrin was used as a PS in cancer PDT trials. For cell experiments, they observed that the effective concentrations of PS-PDT ranged from 0.1 to 0.8 μg.mL^−1^. As for the in vivo efficacy in tumor models of esophageal cancer and hepatoma, the concentration of PS-PDT was maintained between 0.5 and 2 mg.kg^−1^. They therefore concluded that PS was able to inhibit tumor growth and was more efficient when compared to Photofrin [63]. The same PS was used by Wu et al. in the in vitro treatment of breast cancer using PDT. They observed that PS was able to induce cellular apoptosis and the collapse of F-actin filaments by ROS production [64].

In 2021, Zhang and colleagues developed a study involving the synthesis of nanocarriers of silk fibroin coated with MnO_2_ for the encapsulation of chlorin e6 (NPs SF@MnO_2_/Ce_6_), with a focus on evaluating the efficiency of the use of PDT in breast cancer. Elevated tumor inhibitory effects were observed both in vitro and in vivo. Therefore, the authors suggested that the results obtained are promising for the use of these nanocarriers in future applications [59].

The phthalocyanine derivatives (PC) were also shown to be efficient PSs in the treatment of cancer, since they can penetrate deeper tissues. However, there are some limitations, such as low water solubility, inefficient selectivity, and removal rate from the biological system. Some studies have focused on the development of nanosystems carrying PCs [70]. In this context of Drug Delivery Systems, aluminum–phthalocyanine chloride associated with poly(methyl vinyl ether-co-maleic anhydride) nanoparticles (AlPc-NP) was developed as a PS for the in vitro treatment of breast cancer using PDT. It was noted that AlPc-Nps demonstrated improved photophysical characteristics in aqueous media and an increased photodynamic effect against breast tumor cells. However, more studies will be needed [60]. Another PDT study was performed through the synthesis of four new axially disubstituted silicon (IV) phthalocyanines (SiPCs). The spectroscopic properties, quantum yields of ¹O_2_, and in vitro photodynamic activities in hepatocarcinoma and gastric cancer cells were evaluated. Two chlorine-containing SiPCs showed much higher photocytotoxicity compared to Ce6. In addition, PSs could bind to mitochondria and cause cell death by apoptosis [61].

Pucelik et al. demonstrated that PDT with pluronic micelles of redaporphyrin can increase the cellular uptake, as well as oxidative stress, when compared to PS alone. They observed that the modification of redaporphyrin in Pluronic micelles was also effective in treating cancer in vivo, since it increased the tumor selectivity and showed greater production of ROS. The application of PDT with a light dose caused a 100% cure of melanoma after one year of treatment [68].

Another class of photosensitizing agents that have been extensively studied for cancer PDT are chemical substances from natural or synthetic sources of organic or inorganic origin [71]. Kumar et al. reported the green synthesis of a series of 2-(3,5-dimethyl-1H-pyrazol-1-yl)-1-arylethanones and tested their potential as active biological agents in several cancer cell lines. They observed that some of the compounds showed greater selectivity for colon and lung cancer strains. They showed that, for the lung tumor lineage, the viable inhibitory effect was similar to the chemotherapy drug carboplatin [54].

Other compounds synthesized following green protocols were described by Panagopoulos et al. Through green microwave irradiation in the presence of Pbr322 plasmid DNA under UVB and UVA and visible light, eleven quinazolinon-4(3H)-ones of anthranilic acids were synthesized. It was observed that the new quinazoline derivative 6-nitro-quinazolinone in combination with UVA irradiation showed a photodynamic effect for different melanoma and glioblastoma cell lines. Substituted quinazolinones may be effective as a new PS for promising biotechnological applications, as well as approaches involving PDT [53].

In this context, in the literature, there are many works that sought to improve mainly synthetic PSs. On the other hand, agents with natural origin have potential for application in PDT and have been extensively explored for the treatment of various types of cancer [72]. Hammerle et al. demonstrated that photoactive pigments could be isolated from mushrooms. They were able to isolate three PSs based on the structure of anthraquinones and tested their antitumor activity in tumor cell lines. The results demonstrated that dimeric anthraquinone (–)-7,7′-biphyscion was able to induce apoptosis under blue light irradiation in lung tumor cell lines. It was concluded that it was the first fungal PS from basidiomycetes belonging to the anthraquinone class. Therefore, it could be considered a potential candidate for application in PDT [52].

Another work based on Hypocrelin A isolated from the fungus reported by Qi et al. could induce apoptosis in lung adenocarcinoma lineage by increasing intracellular ROS. They concluded that further studies with Hypocrelin A would be needed, since this compound showed photodynamic action in tumor cell lines and a high yield of ¹O_2_ [62]. Traditional Chinese medicines (TCM) have been widely used as photosensitizing agents [72]. In this sense, the group by Wu et al. published two studies involving PSs isolated from a traditional Chinese medicine rhizome for application in PDT of two types of cancer. In the first study, they evaluated the photodynamic activity of palmatin hydrochloride (PaH) in colon adenocarcinoma cell lines. In the second study, the evaluation was performed on a breast cancer cell line. They observed that light dose-dependent phototocytotoxicity in the colon tumor cell lineage, and early and late apoptosis rate was noted at a concentration of 5 μmol.L^−1^ and an energy density of 10.8 J.cm^−2^. For the breast cancer cell line study, they noted that the concentration required to induce apoptosis was less than 0.087 μmol.L^−1^. They concluded that natural PS showed potential for PDT in breast cancer and colon adenocarcinoma [65,66].

Hypericin belongs to the perylenequinone class and is a natural PS widely explored for antitumor PDT. It demonstrates such favorable characteristics as the absorption wavelength in the electromagnetic spectrum range, low toxicity, and high quantum yield of triplet oxygen conversion to a singlet state and the formation of other radicals such as superoxide anions [64,67,71]. Gonçalves et al. showed that hypericin could effectively kill laryngeal cancer tumor cells with a long incubation time and low dose of PS [67].

The use of PDT in the treatment of cancer presents several challenges, mainly in regard to the development of efficient PS that are nontoxic or that have reduced the toxicity for clinical application. Therefore, preclinical studies are essential to understand parameters such as light doses, PS concentration, exposure time, and light delivery mode [71,73,74].

In this sense, a better understanding of the efficiency of the different PSs used in PDT in a clinical application setting is demonstrated in Table 2. Most of the widely used PDT protocols approved for cancer, particularly for non-melanoma skin cancer are based on PS 5-aminolevulinic acid (ALA) and its methyl ester (Metvix) [50]. However, these studies were not detailed in this review.

Some of the PSs found are in ongoing clinical trials for cancer treatment. The safety and tolerability of PS disulfonated tetraphenyl chlorin (TPCS2a) was evaluated in 22 patients with advanced-stage solid tumors over a period of five years. PDT resulted in adverse effects such as photosensitivity on the skin in the region of application, pain, and respiratory failure. The maximum tolerated dose was 1.0 mg.kg^−1^, and the treatment dose was 0.25 kg.mg^−1^ for further studies, and no deaths related to photochemical internalization treatment were reported [77].

2-(1-Hexyloxyethyl)-2-devinyl pyropheophorbide-a (HPPH) is a chlorine-based PS that exhibits improved photophysical and pharmacokinetic characteristics compared to chlorine. It was demonstrated from a study involving 29 patients that it is safe to treat early stage laryngeal cancer [79].

Santos et al. conducted a study involving the use of Redaporfin and an anti-PD1 antibody designed to reduce photosensitivity, reach deeper tissues and, consequently, be more effective and safer compared to other PSs approved for clinical use. For this, a patient with a head and neck tumor underwent treatment with PDT. They observed that Redaporphyrin-PDT was able to destroy all visible tumors [80].

In this case, the authors described the results of silicon (IV) phthalocyanine (SiPC) for nonmelanoma skin cancer. A group of 43 patients underwent SiPC-PDT and were followed up to 14 days. They did not observe significant local toxicity or photosensitivity. The clinical response can be correlated with Pc-4-PDT-induced apoptosis analyzed by the increase in caspase-3 in cutaneous lesions [81].

Among the PS approved for PDT for cancer, Talaporfin (TPS) for bile duct carcinoma (CBD) was highlighted. In 2012, Nanashima et al. published a preliminary clinical study involving seven patients with CBD. Promising results were seen, as three patients remained cancer-free for a period of 6–13 months. In addition, they concluded that TPS-PDT improved the patient’s quality of life compared to conventional PDT. In 2019, Nanashima et al. described a new study involving two patients with CBD. In one of the patients, mild sensitivity occurred, and the end of the treatment finished with no reports of side effects. However, after 155 days of PDT, the patient died. In case 2, no side effects were observed, but on day 132, the patient died. They concluded that long-term survival and quality of life were not achieved. Therefore, they suggest that PDT chemotherapy or brachytherapy needs to be further studied clinically [75,76].

Ji et al. conducted a study employing another chlorin-e6-derived PS for the treatment of lung cancer patients. Radachlorin^™^ is a second-generation PS with promising characteristics for PDT. The authors evaluated ten patients with obstructive advanced non-small-cell lung cancer. They observed that 20% of the patients showed satisfactory results, 70% showed partially successful results, and 10% (one patient) showed unsatisfactory results with a response that had no visible distal bronchus. Other major complications were not observed. In addition, eight patients are undergoing another treatment, and the one-year survival rate after PDT was 70% [78].

## 5. Conclusions

Significant progress has been made in obtaining, characterizing, and applying PSs in cancer PDT. In this review, the application of second and third generation PSs were mainly elucidated, since they can reach deeper tissues, and they overcome several limitations associated with first-generation PSs. We updated and presented studies that used PDT in the treatment of cancer in preclinical studies, aiming at its eventual use in clinical application. In addition, we address some studies that have already shown promising results with the use of PSs in antitumor therapy in clinical practice.

Finally, considering the PS potential of some biomolecules produced by plants so far, the biotechnological route appears as a promising alternative for the application of these biomolecules in PDT. However, several comparative studies are still needed, not only to evaluate the potential of microbial based-PSs in relation to their stability and safety but also to provide more information regarding the processing parameters from upstream and downstream to obtain these biomolecules. Thus, following the results reviewed, we recommend the scientific community that works with PDT and the production of PSs to implement studies concerning the effectiveness of microbial-based PSs in PDT in order to provide the required scientific information for society in general.

## 6. Challenges and Future Trends

In the last few years, PDT has attracted a lot of attention as a promising method in cancer treatment, due to its innovative advantages, being a noninvasive procedure causing little damage to organs and tissues and being unrelated to multidrug resistance [82]. The most used treatments include surgery, chemotherapy, and radiation therapy, methods that result in severe side effects, such as loss of smell and blurred vision. Therefore, PDT becomes very promising, inducing the production of cytotoxic/reactive oxygen that leads to the apoptosis of cells without causing damage to normal tissues [83].

However, traditional PDT faces some challenges when applied to cancer treatment, such as non-selectivity and nonspecific PS distributions, insufficient activation of PSs within the tumor cell microenvironment, and reduced light penetration into deeper tissues, thus decreasing the treatment efficacy [84]. Therefore, in order to bring improvements to traditional PDT, it is possible to associate it with the use of nanotechnology, where limitations related to PSs, such as hydrophobicity, which results in aggregation in biological media, leading to low bioavailability, would be reduced. Additionally, active targeting nanoparticles are being highly studied as promising ideas for uses associated with PDT, ensuring effective protection and the targeting of drugs. A study was performed with cellular microvesicles, where the hydrophobicity of PSs is resolved by passive diffusion using a natural, cell-derived system [84]. A major issue related to the use of nanomaterials is biosafety, requiring a greater understanding of toxicity after light irradiation and in vitro and in vivo studies, and evaluating the means of excretion of the material by the body [85].

In this context, in order to minimize these limitations related to PDT, it is possible to investigate more effective PSs with targeted action on cancerous tissues. PSs of natural/biotechnological origin have been gaining due attention in recent years, where several studies have presented numerous applications related to photodynamic reactions [86]. Studies by Zhang et al. reported the use of a PSs that combine a copolymer type with pyropheophorbide (Ppa), a derivative of chlorophyll, naturally occurring photosynthetic pigments. The PS was given the name POEGMA-b-P (MAA-co-VSPpaMA), exhibiting ROS release properties in a smart way when bound to NPs, dissolved by cancer cells and functionalized with reactive oxygen reactive vinyl thioether. Additionally, the ligand used with the copolymer is sensitive to ROS, facilitating bond breaking when the molecule is generated by the reactions [87].

Thus, it is noticeable that there are multiple possibilities for better application and obtainment of PSs as therapeutic agents through PDT. However, what proves to be the biggest challenges when aiming for a clinical application would be a better screening of agents and standardization of the numerous associated approaches. The relative lack of understanding of the structural determinants of PSs and how they contribute to the phototoxic process makes it very difficult to develop better PSs through recombinant technology, which makes conventional PSs the most palpable options, but at the same time, they lack an effective screening method in order to elect the most suitable one for a particular application [12]. The associated techniques for performance improvement also represent a challenge, because if the chosen technique was the coating with cell membranes of the nanostructures carrying PSs for better targeting and bioavailability, problems such as the need for autologous cells for coating the nanostructured system, generating challenging demands for large-scale production and quality maintenance, besides becoming a personalized therapy—in other words, depending on the chosen technique to act synergistically with the PSs—other standardization challenges will arise [10].

## Figures and Tables

**Figure 1 molecules-27-06848-f001:**
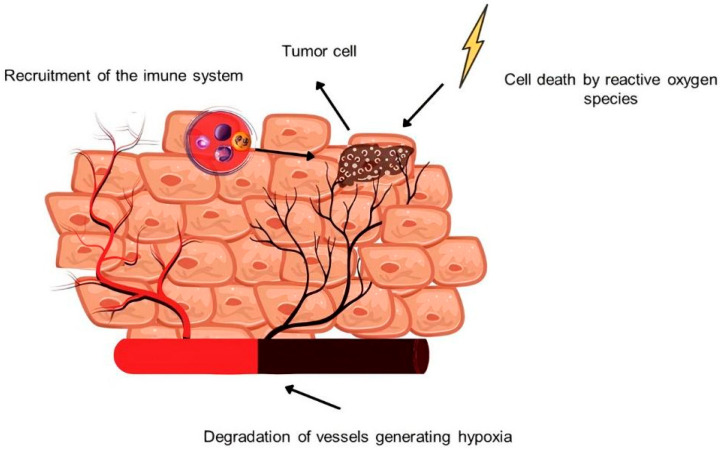
Three different mechanisms of PDT to combat tumor cells, being the process of direct cytotoxicity, inflammatory processes modulated by activation of the immune response, and hypoxia by vessel occlusion. Source: Authors.

**Figure 2 molecules-27-06848-f002:**
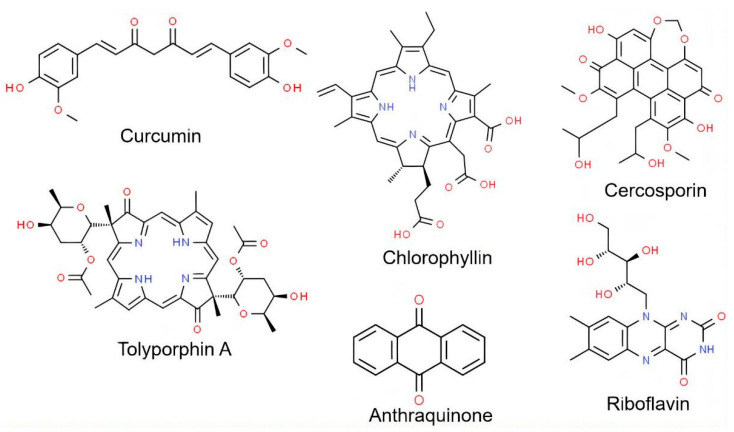
Chemical structure of the main biotechnological photosensitizers applied in PDT. Source: Authors.

**Figure 3 molecules-27-06848-f003:**
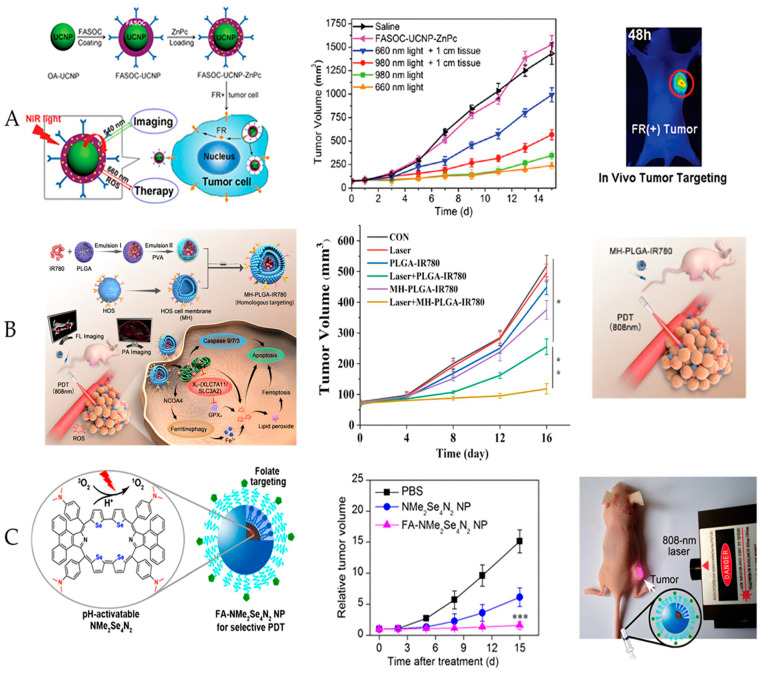
Schematic illustration of the main studies involving new types of PS for application in cancer PDT. (**A**) Synthesis of the FASOC-UCNP-ZnPc nanoconstruct and folate-mediated binding of tumor cells with folate receptor expression. Comparison of the therapeutic efficacy of deep tissue PDT triggered by 980 and 660-nm light. (**A**) Tumor growth of mice in different treatment groups within 15 days. Adapted from [51]. Reproduced with permission from the American Chemical Society, Copyright © 2013. (**B**) Construction of the MH-PLGA-IR780 NPs and the specific killing mechanism of the targeted theranostic nanoplatform-mediated PDT approach. The tumor volume was measured during the therapeutic period. (The data are presented as the mean ± SD values; *n* = 5, * *p* < 0.05, and ** *p* < 0.01.) Adapted from [58]. Reproduced with permission from Springer Nature, Copyright © 2022. (**C**) Targeted PDT on a Hela tumor-bearing mouse intravenously injected with FA-NMe2Se4N2 NPs. Change of relative tumor volume (V/V0) after mice were intravenously injected with PBS, NMe2Se4N2 NPs, or FA-NMe2Se4N2 NPs and irradiated with a 808-nm laser at 100 mV cm^−2^ for 30 min. Data are the means ± SD (6 mice per group), *** *p* < 0.001 compared to other groups using a one-way ANOVA. Adapted from [57]. Reproduced with permission from the American Chemical Society, Copyright © 2013.

**Table 1 molecules-27-06848-t001:** Overview of photosensitizers used for in vitro/in vivo treatment of cancer in the last decade.

Photosensitizer (s)	PS Class	Cancer Type (s)	Application	PS-PDT Effective Concentrations	Ref.
Upconversion nanoconstruct—targeted with folate-modified amphiphilic chitosan loaded with zinc(II) phthalocyanine (FASOC-UCNP-ZnPc)	Phthalocyanines Derivatives	Hepatocellular carcinoma and sarcoma	In vivo	<150 mg.kg^−1^ triggered by 660 and 980 nm light (0.2 W.cm^−2^, 30 min)	[51]
Dimeric anthraquinone (–)-7,7′-biphyscion	Anthraquinones Derivatives	lung cancer, cervical cancer, stomach cancer and urinary bladder carcinoma	In vitro	0.064 µmol.L^−1^ combined with 9.3 J.cm^−2^ light dose (λexc = 468 nm)	[52]
6-Nitro-Quinazolin-4(3H)-one	Quinazoline Derivatives	glioblastoma and melanoma	In vitro	50 μmol.L^−1^ and then irradiated at 365 nm using a UVA lamp for 1 and 2 h.	[53]
2-(3,5-dimethyl-1Hpyrazol-1-yl)-1-arylethanones	Pyrazole Derivatives	colon cancer, prostate cancer, ovarian cancer, and lung cancer.	In vitro	Not reported.	[54]
Hf–porphyrin nanoscale metal–organic framework, (DBP–UiO)	Porphyrin derivative	head and neck cancer	In vitro/in vivo	5–100 μmol.L^−1^ and 3.5 mg.kg^−1^(in vivo) combined with 90–180 J.cm^−2^ light dose (100 mW.cm^−2^, 15 and 30 min).	[55]
Cationic porphyrin-cisplatin conjugate (Pt-1)- polymeric nanoparticles (NP@Pt-1)	Porphyrin derivative	colon carcinoma	In vitro/ in vivo	0.025–20 μmol.L^−1^ and 3.5 mg.kg^−1^(in vivo) combined with 6.95 J.cm^−2^ light dose (100 mW.cm^−2^, 15 and 30 min).	[56]
Selenium-rubyrin (NMe_2_Se_4_N_2_)-loaded nanoparticles functionalized with folate (FA)	Transition metal complex	cervical carcinoma	In vitro/ in vivo	35 μg.mL^−1^ and 0.5 mg.kg^−1^(in vivo) combined with 30 J.cm^−2^ light dose (635 nm and 808 nm laser, 100 mW.cm^−2^, 30 min).	[57]
Constructed homologous targeting-based nanoplatform (MH-PLGA-IR780 NPs)	NIR-absorbing PSs	osteosarcoma	In vitro/in vivo	5 μg.mL^−1^, 808 nm laser, 1 and 1.5 W.cm^−2^.	[58]
MnO_2_-capped silk fibroin (SF) nanoparticles with chlorin e6 (Ce6) encapsulated	Chlorin	breast cancer	In vitro/ in vivo	40 μg.mL^−1^ and 1 mg.kg^−1^ (in vivo), 808/660 nm laser, 1.5/1.0 W.cm^−2^.	[59]
Aluminum-phthalocyanine chloride associated with poly(methyl vinyl ether-co-maleic anhydride) nanoparticles	Phthalocyanines Derivatives	breast cancer	In vitro	0.25 μmol.L^−1^ combined with 3.82 J.cm^−2^ light dose	[60]
Silicon (IV) phthalocyanine (SiPC)	Phthalocyanines Derivatives	hepatocarcinoma and gastric cancer	In vitro	9 nmol.L^−1^ to 33 nM combined with 27 J.cm^−2^ light dose (λ > 610 nm, 15 mW.cm-^2^, 30 min).	[61]
Hypocrellin A (HA)	Natural hypocrellins	lung adenocarcinoma	In vitro	0.08 μmol.L^−1^, 470 nm LED light irradiation.	[62]
Sinoporphyrin Sodium	Photofrin	eleven human cancer cell line	In vitro/ in vivo	0.1–0.8 μg.mL^−1^, 630 nm laser at fluence rate of 30 mW.cm^−2^ total illumination power 5.4 J.well^−1^. 0.5–2 mg.kg^−1^, fluence rate of 127.7 mW.cm^−2^ total illumination power 60 J/animal.	[63]
Sinoporphyrin Sodium	Photofrin	breast cancer	In vitro	2–4 μ.mol.L^−^ at fluence rate of 23.85 mW.cm^−2^ combined with 5.72 J.cm^−2^ light dose.	[64]
Palmatine hydrochloride (PaH)	Quinolone-based alkaloids	breast cancer	In vitro	0.087 μ.mol.L^−1^, 470 nm LED light irradiation, combined with 10.8 J.cm^−2^ light dose.	[65]
Palmatine hydrochloride (PaH)	Quinolone-based alkaloids	colon adenocarcinoma	In vitro	5 μ.mol.L^−1^, 470 nm LED light irradiation, combined with 10.8 J.cm^−2^ light dose.	[66]
Hypericin	Perylenequinone/Natural PS	larynx carcinoma	In vitro	11 and 25 nmol.L^−1^, combined with 6 and 12 J.cm^−2^ light dose.	[67]
Redaporfin-P123 micelles	bacteriochlorin derivative	melanoma	In vitro/ in vivo	5 μ.mol.L^−1^, 735 nm LED light irradiation, combined with 0.97 J.cm^−2^ light dose. 1.5 mg.Kg^−1^, 750 nm LED light irradiation, combined with 74 J.cm^−2^.	[68]

**Table 2 molecules-27-06848-t002:** Photosensitizers in ongoing and/or approved clinical trials for PDT for cancer.

Photosensitizer (s)	PS Class	Cancer Type (s)	PS-PDT Effective Concentrations	Ref.
Talaporfin, mono-L-aspartyl chlorin e6, NPe6, LS11 (Laserphyrin)	chlorin(e6) derivative	bile duct carcinoma	40 mg.m^−2^ combined with 100 J.cm^−2^ light dose (664 nm, 4–6 h).	[75]
Talaporfin, mono-L-aspartyl chlorin e6, NPe6, LS11 (Laserphyrin)	chlorin(e6) derivative	bile duct carcinoma	40 mg.m^−2^ combined with 100 J.cm^−2^ light dose (664 nm, 6 h).	[76]
Disulfonated tetraphenyl chlorin (TPCS2a)	chlorin	Solid tumor	0.25 mg.kg^−1^, fluence rate of 100 mW.cm^−2^, 60 J.cm^−2^ light dose (652 nm, 4–6 h).	[77]
Radachlorin	chlorin derivative	obstructive advanced non-small-cell lung cancer	1 mg.kg^−1^ combined with 200 J.cm^−2^ light dose (662 nm, 11 min 6 s).	[78]
2-(1-Hexyloxyethyl)-2-devinyl pyropheophorbide-a (HPPH)	chlorin derivative	laryngeal cancer	4 mg.kg^−1^ at fluence rate of 100 mW.cm^−2^ combined with below 100 J.cm^−2^ light dose (665 nm).	[79]
Redaporfin, LUZ11 or F-2BMet, 5,10,15,20-tetrakis(2,6-difluoro-3-N-methylsulfamoylphenyl)-bacteriochlorin	bacteriochlorin derivative	head and neck squamous carcinom	0.75 mg.kg^−1^combined with 50 J.cm^−2^ light dose (749 nm).	[80]
Silicon (IV) phthalocyanine (SiPC)	Phthalocyanines Derivatives	non-melanoma skin cancer	0.1 mg.mL^−1^ combined with 100–150 mJ.cm^−1^ light dose.	[81]

## Data Availability

Not applicable.

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
