# Peer review of "Development of Biotechnological Photosensitizers for Photodynamic Therapy: Cancer Research and Treatment—From Benchtop to Clinical Practice"

_molecules, 2022, doi:10.3390/molecules27206848_

Round 1

Reviewer 1 Report

The authors has written a review work on photosensitizers and their implement in clinical use. They have described the most important conditions for a photosensitizers to be applied in PDT, taking into account also those which are from microbial origin or lad-made.

The manuscript contains many examples, however, at some point it is difficult to follow. My main criticism is that there are many detailed works but with a few connection between them, mainly in the last pages. I would suggest describing those examples with a more connected message, meaning that, what is improved from one example to the other, and discuss better the improvements in the field. In this context, the conclusion section is quite brief.

There are other few comments to take into account:

-        there are sometimes that the called to photosensitizers as PS but others as FS.

-        Page 2, lines 60-66. This sentence is not clear. They make differences between the wavelength to track and the wavelength to treat, but not in a clear way

-        page 5, lines 189-201. This paragraph is repetitive. Similar conditions as already described in page 4 lines 132-144. Please unify them

Author Response

Reviewer(s)' Comments to Author: 

Reviewer: 1 

The authors has written a review work on photosensitizers and their implement in clinical use. They have described the most important conditions for a photosensitizers to be applied in PDT, taking into account also those which are from microbial origin or lad-made.

The manuscript contains many examples, however, at some point it is difficult to follow. My main criticism is that there are many detailed works but with a few connection between them, mainly in the last pages. I would suggest describing those examples with a more connected message, meaning that, what is improved from one example to the other, and discuss better the improvements in the field. In this context, the conclusion section is quite brief.

Authors´ Response: Thank you for this valuable feedback. In our paper, we summarize recent studies on third-generation PS, mostly and at an early stage, for different types of cancer. Therefore, for the most part, it would not be possible to make a fair comparison between the results obtained. The authors demonstrated the use of PDT in the treatment of cancer presents several challenges, mainly in regard to the development of efficient PS that are non-toxic or that have reduced toxicity for clinical application. Therefore, pre-clinical studies are essential to understand parameters such as light doses, PS concentration, exposure time and light delivery mode [72,74,75]. We have added a new  figure with the illustrative scheme about the most innovative studies and a topic on future prospects and challenges to clarify this.

There are other few comments to take into account:

-        there are sometimes that the called to photosensitizers as PS but others as FS.

Authors´ Response: As suggested, the change was made, highlighted in the text.

-        Page 2, lines 60-66. This sentence is not clear. They make differences between the wavelength to track and the wavelength to treat, but not in a clear way.

Authors´ Response: As suggested, the change was made, highlighted in the text.

-        page 5, lines 189-201. This paragraph is repetitive. Similar conditions as already described in page 4 lines 132-144. Please unify them

Authors´ Response: As suggested, the change was made, highlighted in the text.

Reviewer 2 Report

The authors have submitted an interesting review article entitled "Development of Biotechnological Photosensitizers for Photodynamic Therapy: Cancer research and treatment - From benchtop to clinical practice "which summarized the recent advances in PDT for cancer treatment. I suggest this article be published after a serious major revision.

Comments:

 1- First of all, I would like to recommend authors to design a better “Graphical Abstract” for this study to better show the whole story in a simple and informative manner. To this end, you can easily use the Biorender website to make a simple, but intuitive and informative illustration.

 2- Please carefully revise the manuscript to remove grammatical errors and vague sentences. Some of the sentences are unnecessarily long (like the very first sentence of the introduction) which makes it difficult and boring for the readers to follow them. Please double-check the whole manuscript and revise all.

3- A good and comprehensive review paper needs at least 3-5 master figures which summarize new findings from impactful papers and 2-3 master Tables which tabulate conditions and results of newly published papers in the field. The authors should add new figures related to how these photoresponsive materials function in terms of PDT. Please read the following Key Papers and many other available ones with a similar concept to try to promote the quality of this manuscript.

“Photosensitizers for Photodynamic Therapy /  https://doi.org/10.1002/adhm.201900132””
“Photodynamic therapy: photosensitizers and nanostructures/ https://doi.org/10.1039/D0QM00922A”
“Internal light source for deep photodynamic therapy / https://doi.org/10.1038/s41377-022-00780-1”

5- Some of the references in the introduction part are too old (e.g., 1996, 2005, 2007, etc.) and it is not acceptable at all. A myriad of research bodies has been published in recent years and you can find similar concepts and cite them in your paper rather than more than 3 decades old references. Moreover, in the introduction part and related to photoresponsive materials and the fundamentals of this field, please read and add valuable information from the following paper as well: https://doi.org/10.3390/ijms23042223 , https://doi.org/10.1016/B978-0-12-818806-4.00010-3

6- It is required to provide a section with a “Future Trend” along with a “Challenges” perspective to point the readers out to what direction this subject is going and what are obstacles in the way and what should be done in the future.

Author Response

Reviewer(s)' Comments to Author: 

Reviewer: 2

The authors have submitted an interesting review article entitled "Development of Biotechnological Photosensitizers for Photodynamic Therapy: Cancer research and treatment - From benchtop to clinical practice "which summarized the recent advances in PDT for cancer treatment. I suggest this article be published after a serious major revision.

 Comments:

1- First of all, I would like to recommend authors to design a better “Graphical Abstract” for this study to better show the whole story in a simple and informative manner. To this end, you can easily use the Biorender website to make a simple, but intuitive and informative illustration.

Authors´ Response: Thank you for this valuable feedback. Regarding the graphical abstract. We summarize in the illustration on obtaining PS of biotechnological origin and the possibility of preclinical and clinical application. Resulting in death of target cells. As suggested, the image resolution has been changed (increased).

2 - Please carefully revise the manuscript to remove grammatical errors and vague sentences. Some of the sentences are unnecessarily long (like the very first sentence of the introduction) which makes it difficult and boring for the readers to follow them. Please double-check the whole manuscript and revise all.

Authors´ Response: We regret there were problems with the English. The paper has been carefully revised by a native English speaker to improve the grammar and readability (Prof. Andy Cumming - UK - english revision service)      . As suggested, the change was made, highlighted in the text.

3 - A good and comprehensive review paper needs at least 3-5 master figures which summarize new findings from impactful papers and 2-3 master Tables which tabulate conditions and results of newly published papers in the field. The authors should add new figures related to how these photoresponsive materials function in terms of PDT. Please read the following Key Papers and many other available ones with a similar concept to try to promote the quality of this manuscript.

Photosensitizers for Photodynamic Therapy/  

https://doi.org/10.1002/adhm.201900132

Photodynamic therapy: photosensitizers and nanostructures https://doi.org/10.1039/D0QM00922A

Internal light source for deep photodynamic therapy

 https://doi.org/10.1038/s41377-022-00780-1

Authors´ Response: As suggested we have added a new figure (Figure 3)      with the illustrative scheme about the most innovative studies. The change was made, highlighted in the text.

4 - Some of the references in the introduction part are too old (e.g., 1996, 2005, 2007, etc.) and it is not acceptable at all. A myriad of research bodies has been published in recent years and you can find similar concepts and cite them in your paper rather than more than 3 decades old references. Moreover, in the introduction part and related to photoresponsive materials and the fundamentals of this field, please read and add valuable information from the following paper as well: 

https://doi.org/10.3390/ijms23042223

https://doi.org/10.1016/B978-0-12-818806-4.00010-3

Authors´ Response:As suggested, the change was made, highlighted in the text (p. 3 line 94-105).

5 - It is required to provide a section with a “Future Trend” along with a “Challenges” perspective to point the readers out to what direction this subject is going and what are obstacles in the way and what should be done in the future.

Authors´ Response: As suggested, the change was made, highlighted in the text (p. 16; section 6.).

Round 2

Reviewer 1 Report

The authors have improved the manuscript. I consider that the current revised version is suitable for publication in Molecules as it is. 

Reviewer 2 Report

The manuscript is well amended and it is ready for publication. The author has covered all my concerns and I have no further comments.